# Patient experience of long-term recovery after open fracture of the lower limb: a qualitative study using interviews in a community setting

Sophie Rees ,[1] Elizabeth Tutton ,[2] Juul Achten,[2] Julie Bruce,[1] Matthew L Costa[2]

SR and ET are joint first authors.

[1]Warwick Clinical Trials Unit, University of Warwick, Coventry, UK
[2]NDORMS, University of Oxford, Oxford, UK

**Correspondence to**
Elizabeth Tutton;
liz.tutton@ndorms.ox.ac.uk

## ABSTRACT

**Objectives** Treatment of open fractures is complex and patients may require muscle and skin grafts. The aim of this study was to gain a greater understanding of patient experience of recovery from open fracture of the lower limb 2–4 years postinjury.

**Design** A phenomenological approach was used to guide the design of the study. Interviews took place between October 2016 and April 2017 in the participants' own homes or via telephone.

**Setting** England, UK.

**Participants** A purposive sample of 25 patients were interviewed with an age range of 26–80 years (median 51), 19 were male and six female, and time since injury was 24–49 months (median 35 months).

**Results** The findings identified a focus on struggling to recover as participants created a new way of living, balancing moving forward with accepting how they are, while being uncertain of the future and experiencing cycles of progress and setbacks. This was expressed through three themes: (i) 'being disempowered' with the emotional impact of dependency and uncertainty, (ii) 'being changed' and living with being fragile and being unable to move freely and (iii) 'being myself' with a loss of self, feeling and looking different, alongside recreation of self in which they integrated the past, present and future to find meaningful ways of being themselves.

**Conclusion** This study identified the long-term disruption caused by serious injury, the hidden work of integration that is required in order to move forward and maximise potential for recovery. Supportive strategies that help people to self-manage their everyday emotional and physical experience of recovery from injury are required. Research should focus on developing and testing effective interventions that provide support and self-management within a holistic rehabilitation plan.

**Trial registration number** Current Controlled Trials ISRCTN33756652; Post-results.

## INTRODUCTION

Open fractures of the lower limb (where the bone protrudes through the skin) occur in an estimated 30% of fractures of the tibia.[1] Treatment is complex as the wound requires surgical cleaning before fixation of the bone,

### Strengths and limitations of this study

► Use of in-depth qualitative interviews has provided rich data and new insights into the hidden work undertaken by patients as they recover from major lower leg injury.
► The variation and patterns within the patients' experience may help healthcare providers focus effective strategies to maximise emotional and physical recovery over the longer term.
► Our study was intentionally exploratory in nature and the resulting framework requires further exploration in diverse samples to assess its transferability.

often followed by muscle and skin grafting. The UK WOLLF Wound management of Open Lower Limb Fractures trial compared standard dressings with negative pressure wound therapy (NPWT).[2] NPWT is a type of dressing that applies a gentle suction to the wound removing excess fluid. Two qualitative substudies of this trial were undertaken to explore the lived experience of patients with an open fracture of the lower limb, first during hospitalisation, and second, this paper presents the findings from 2 to 4 years postinjury.

There is increasing evidence that the impact of open fracture of the lower limb can be life changing with prolonged periods of recovery. Embodied vulnerability conveys how patients with open fractures of the lower limb struggled to cope with their emotional distress, their changed body which included their wounds, body image, pain and the uncertain nature of their future life.[3] Recovery can be slow and patients can find it hard to return to their preinjury lives even 2 years after an open fracture.[4] Studies with a broader sample of injured patients identify the ongoing challenge of daily living, being able to work and body image.[5] There is also

evidence of persisting psychological distress.[6] In order to provide direction for long-term support and rehabilitation, this qualitative study develops existing knowledge by exploring the research question, what are patients' experiences of recovery from an open fracture of the lower limb 2—4 years postinjury?

## METHODS

This study was informed by phenomenology and the work of Heidegger[7] used in other studies of injury.[8 9] This enabled participants to share their experience of everyday life within their social and cultural context. It uncovered insights into how they know and understand their world and what is important to them. Immersion in the data and interpretation led to an understanding of the meanings inherent in the everyday world of the participant.

A purposive sample of 25 participants were recruited between October 2016 and April 2017, during routine follow-up. Interviewees were participants in the UK WOLLF trial[2] and had received reconstructive surgery for an open fracture to the lower limb. Purposive sampling included a range of sex, age and grade of severity of injury. A key eligibility criterion for the UK WOLLF trial was that at the end of the first surgical wound debridement, the wound could not be closed. This was required for the trial, as the NPWT dressings under investigation cannot be applied to closed wounds, but does not fit clearly with the existing classification systems. Only patients with a higher grade of open fracture were included where skin or muscle grafts are required. Three subdivisions of the Gustilo and Anderson[10] classification were used, grade II, grade III (inclusive of a/b) or grade IIIc with a vascular injury. See table 1 for information on the participants' sex, age, time since injury, cause of injury and injury severity. Two people approached chose not to take part due to personal circumstances and lack of time. Respondents received a patient information sheet and provided written or oral consent to take part in an interview. The interviews focussed on the participants' lived experience of recovery from injury and were lightly structured to cover their thoughts, feelings, activities, function and work. Open questions such as 'what has it been like for you since your injury', with prompts such as 'tell me more about that', 'how did that feel', were used to enable participants to share their experience. Two female health researchers with PhDs who did not know the participants undertook the interviews (SR (n=15) and ET (n=10)), either face-to-face (usually at the participant's own home, once at a nearby hospital) or over the telephone. Two participants were accompanied by their partners. Care was taken to support participants and respect their privacy and dignity. One interviewer had experience of interviewing patients with cancer and a background in medical sociology, the other had prior experience of interviewing patients with traumatic orthopaedic injuries. Interviews lasted 25–100 min (average 60 min) and were digitally audio-recorded and transcribed verbatim. Data

**Table 1** Information about the participants

| Characteristics | Number of participants |
|---|---|
| Sex | |
| Male | 19 |
| Female | 6 |
| Age (years) median 51, range 26–80 | |
| 18–29 | 4 |
| 30–49 | 8 |
| 50–69 | 9 |
| 70–80 | 4 |
| Time since injury (months) median 31, range 24–49 | |
| 24–35 months | 14 |
| 36–49 months | 11 |
| Cause of injury | |
| Car collision | 8 |
| Motorbike collision | 5 |
| Pedestrian–vehicle collision | 3 |
| Fall from height/stairs | 3 |
| Crush injury at work | 3 |
| Low energy fall (from standing) | 2 |
| Bicycle–vehicle collision | 1 |
| Injury severity, Gustilo and Anderson classification | |
| II | 4 |
| III (inclusive of a/b) | 18 |
| IIIc | 3 |

were anonymised and held on secure password protected University computers. None of the participants wished to see a copy of their transcript.

### Patient and public involvement

Patient and public involvement (PPI) was integral to the design and conduct of the UK WOLLF study. Four patients with similar injuries who were members of the UK WOLLF PPI group helped the researchers reflect on their interpretations during analysis. The UK Musculoskeletal Trauma PPI group are involved in dissemination of the findings.

### Analysis

Analysis was undertaken by two researchers (SR and ET), was an iterative process involving initial coding of sections of the data to label the underlying meaning or 'what is going on' in the data such as 'feeling sad due to lack of progress'. Codes were collected together with other similar codes to create categories such as 'being uncertain'. Differences and challenges within interviews and across interviews were written up in field notes and memos, and discussed. Categories were collected together to create themes or 'structures of experience'[11] such as

'being disempowered'. NVivo V.11 a qualitative software package was used to manage the data. The findings were shared with the broader research team. Differences in interpretation were discussed as part of the process of analysis but in general there was agreement about the nature and content of the three themes.

Rigour was demonstrated through trustworthiness.[12] Both researchers were engaged with the data, held regular discussions and reflected on their positionality throughout analysis. Auditability was demonstrated through the identification of themes and categories and use of quotes to illustrate them. Saturation of themes and categories was achieved. This was indicated when no new themes or categories developed after 18 patients were interviewed. Interviewing continued to ensure that the sample was purposeful and there was a range of codes in each category. Resonance with the findings was identified by four patients from the UK WOLLF PPI group. It was noted that they placed emphasis on existing codes that identified challenges with intimacy and gratefulness for care provided. A workshop including a range of multi-disciplinary staff representing nursing, physiotherapy, trauma surgeons, plastic surgeons and psychologists suggested that the findings reflected their experience of listening to patients in clinical practice, although they held differing perspectives on the degree of importance of items and labelling of codes. For example, they felt some codes could be drawn together under the broader labels of anxiety or depression.

## RESULTS
### Overarching theme: struggling to recover
The findings convey the overarching theme of 'struggling to recover' as an experiential process in which participants aim to make sense of their injury, balance striving to move forward with accepting how they are, and find meaningful ways of living while being uncertain of the future and experiencing cycles of progress and setbacks.

In struggling to recover participants conveyed that their 'taken for granted' ways of living experienced prior to injury were replaced with daily challenge as they endured symptoms such as persistent pain. They often hid the struggle to keep going to maximise participation and maintain a sense of progression.

> They can see people walking but there is no light or pain-meter on top of the head that says 'this guy is in absolute agony but he's not going to stop walking because he doesn't want to go in a mobility scooter'. People look at you and say 'You're getting on great aren't you?' All you really just want to say to them is 'Every day I struggle' and some days you just want to sit and cry your eyes out. (P13)

Loss caused by injury led participants to renegotiate how they live and to integrate their past and present self into new ways of being. This was portrayed through

the themes of being disempowered, being changed and being myself.

### Theme 1: being disempowered
Being disempowered conveyed the emotional and physical impact on participants of loss of personal control over their life and was expressed through the categories 'being dependent on others' and 'being uncertain about the future'.

#### Being dependent on others
Being dependent on others was a process of accepting help from others while striving to maintain a sense of independence through activities that supported their mental and physical well-being. Many participants had prolonged periods where they were not able to put weight through their injured leg, and some had injuries to both legs. Dependency on others created frustration, boredom, distress, dismay, was undignified, lowered their mood and was epitomised by not even being able to perform simple tasks, such as to make a cup of tea.

> I don't know if it's a man thing or male thing, I don't know. It's your dignity, I mean the last person who wiped my arse was my mum and that's it, or my dad, but being a grown man using a bed pan because I couldn't get out of bed was awful. (P06)

In this state of profound dependency, their body became central to everyday life, its needs and limitations explicitly governing every aspect of everyday life.

> You feel a bit like a passenger in it all because you're on the outside looking in and you think you're being a bit of a burden on everyone. At the time I found it quite hard to almost tell people that's how I felt, I feel this, I feel a bit worthless. (P18)

Constant support for daily bodily needs could lower their mood so that they could no longer see the 'light at the end of the tunnel' (P17) evident in suicidal thoughts.

> I thought about topping myself because it's like I've been independent all my life. … One night I had all the tablets, Tramadol. (P06)

Dependency on others was a disempowering experience that led to emotional vulnerability. Expressing how they felt was challenging in the context of being grateful for the care they had received and some had suicidal thoughts.

#### Being uncertain about the future
Being uncertain about the future conveyed the emotional impact of not knowing their potential degree of recovery, and what life would be like in the future. Being uncertain about their future added to participants' sense of disempowerment. Participants found it hard to imagine walking properly or returning to preinjury activities and a 'wait and see' approach added to their sense of uncertainty.

I would say that was the toughest part, was just the un-endingness of it, it was just constant and it was horrible. (P19)

I did think in my mind you know 'it will heal and then I've got rehabilitation' but this whole non-union, it just dragged on and on and on, I couldn't see an end. … I wished I'd known about the non-union, err, it was almost hidden from me. … I said 'well I thought that the operation was successful?' then he told me 'well it was because we would've cut your leg off years ago'. (P21)

There was a degree of anxiety regarding their future and ability to live and work productively.

You want to live for the future now because that's what you've got in front of you but you worry about what it's going to bring, not only physically but financially as well. You have no idea what your pay-out is going to be and so my life is in somebody else's hands. There is that horrible thought that you may have to go back out to work, force yourself to work because financially you can't exist for the rest of your life. (P13)

Their capacity to heal often determined their future; leaving them feeling uncertain, disempowered and anxious. This was exacerbated by setbacks such as non-union (also noted in recreating me) that reduced their ability to sustain recovery.

### Theme 2: being changed

Being changed identified how their body no longer looks, feels or functions in ways it had previously. Everyday life involved a process of renegotiating where participants learnt through experience what they could do and how they felt about their body while striving to regain and retain normal activities. It was expressed through the categories of being fragile and being unable to move freely.

### Being fragile

Being fragile conveyed a sense of their body as looking and feeling less robust, less trustworthy, less reliable and active engagement was required to reconnect in a positive way with their changed body. Participants were predominantly young or middle-aged adults, and many had been previously fit and active individuals. They now felt their body was older, weaker and more fragile.

I can't dig. … I feel like I've got a weakness which has left me feeling like I've got a bit of a disability. (P21)

Participants felt a need to protect their injured body, or at least the injured limb, including avoiding activities due to the fear of experiencing another injury.

You're frightened it's gonna (going to) break again. (P02)

The sense of a whole, continuous body which the participants felt they could trust to do the activities they wanted

to do was disturbed and instead participants could experience their bodies as alien and fragile.

I do go running and I wear long running trousers because I think if I see it (the muscle graft) it makes me feel like there's a weakness there and it makes me very aware, I don't think it is any weaker, the leg, but mentally it makes me think it's a bit weaker and I just find it distracting as well if I exercise it. I probably would be over-cautious and not really exercise it properly if I could see it I think. (P14)

Some participants described feeling distanced or alienated from the injured part of their body.

I don't look at it as my leg anymore. … It is like the legs belongs to somebody else they don't particularly belong to me. (P19)

Their injured body part was no longer fully integrated into their perception of their body. Parts of their body had literally shifted places as skin/muscle from other areas of the body had been transplanted to cover the defect in the leg. Pain and numbness could increase this sense of alienation.

Participants hid or covered up their injured leg, to avoid others perceiving them as fragile, less capable or weak. For some participants, the struggle with healing, pain and lack of mobility was so exhausting, that they thought amputation might improve their chances of mobility and a better quality of life.

There were a couple of times in it all when I thought, would it be better off to say take it off but I knew how hard and how much effort people had put in to make sure that leg was staying where it was. (P18)

If you took me back in time and I knew then what I know now I would have said 'take my leg off now'. I'd be four years down the line with a prosthetic and I could probably still be working. (P13)

Being fragile highlighted the unreliability of their body and continued vulnerability to further injury. The sense of a 'whole' body could be disrupted and amputation was considered in an effort to improve chances of recovery.

### Being unable to move freely

Being unable to move freely was the loss of ease, fluidity and previously taken-for-granted ways of moving, it affected their ability to use certain geographical spaces and effort was required to improve levels of physical activity. Despite being between 2 and 4 years postinjury, participants felt they were not able to move with the same fluidity or spontaneity. The way they walked, their pace, gait or balance was different.

I can't move fast anymore. (P17)

My balance was all over the place, it still is, sometimes I feel like I'm falling to the side, it's a weird feeling. (P01)

I waddle, I walk like a penguin it's the only way I can balance, sort of putting weight on one foot and then going over to the next foot, I can't stride out anymore. (P02)

Specific activities such as being able to kneel prevented them from moving as freely. Some participants felt the area where they had a soft tissue/muscle flap used in reconstructive surgery was heavy (notable when sleeping), or they experienced swelling and pain. Persistent pain reduced their ability to move and join in activities, it varied in type and duration but sometimes was prolonged and often unpredictable.

I sit down for an hour and I get up, oh the pain you can't describe it, but I mean it's for seconds but it's enough to, it's really, really painful. (P02)

Participants experienced restrictions on the spaces and places they felt confident occupying. They were worried about falling and injuring themselves again. The terrain underfoot, types of floor or ground could cause pain leading them to be constantly vigilant.

Participants felt changed, even if they had returned to work, often with an increased sense of fragility, loss of fluidity and pace of movement, combined with specific functional losses. They hid this fragility from others, undertook additional planning for daily activities, limited social participation and changed their work roles in order to cope with their injury.

### Theme 3: being myself

Being myself was an active process of integrating injury into their sense of who they were as a person, bringing together the past, present and future. It was expressed through loss of self and 'recreating me'. Participants worked to integrate their injury into their lives expressed through 'loss of self' and 'recreating me'.

#### Loss of self

Loss of self was expressed as participants felt and looked different and were unable to fulfil their normal roles and activities as they did prior to their injury. In loss of self, there was a sense that the body 'restricts me from being me' and participants had to adapt to being different from their preinjury state.

It seemed like a part of my life stopped at 6.45 on (date of injury) and a new life started. … I used to think nothing of walking ten to fifteen miles a day along disused railway lines and things like that, we were avid walkers. (P04)

Participants' current experience contradicted their memories of who they were.

I have 55 years' worth of memories inside my head so you think 'Oh I used to go running on a Saturday morning'. … You try and be the person you were before and you can't be because you are 95% of what

you used to be. But it's just that 5% area that causes you 100% of the problems. (P13)

Some days I, I could scream. I wake in the morning and 'cause you forget', you know when you go to sleep, and I wake up in the morning thinking I can just jump out of bed and I can't and then it hits you and you think 'Oh God I can't do this!' (P02)

Their body thwarted their attempts to return to their former selves. Despite regaining their independence and feeling that their healthcare team considered them to be recovered, the participants were continually reminded of their injured bodies.

It's very difficult to have a day where you are not conscious of 'that hurts'. … In an active way you are kind of constantly reminded because it's never quite the same day to day. (P22)

In contrast to younger patients, older participants located the injury within the context of prior conditions and they did not experience the same degree of challenge to their identity.

The event (which) bears more on how I think about life in general is having been fortunate to survive a heart attack ten years ago. … So that sort of in a way puts things in perspective. I know falling off a roof can initially be thought of as a life-threatening event but a cardiac arrest is definitely a life-threatening event! (P08)

Participants struggled with the loss of their preinjured self and the activities that defined them. Despite attempts to move forward and integrate with the present, they found their unreliable body provided sensory reminders of their loss and their injury.

#### Recreating me

Recreating me referred to the ways in which participants made sense of their altered selves, worked on their body and mind to find meaningful ways of being and living. In making sense of their injury, they reflected on how life had been and tried to reconcile this with how they felt now.

It did come as a revelation about three weeks ago, just you know, I can't carry on doing this, I can't keep pushing to get my life back, it's got you know, I've got to change, you know it is a bit of a shock! (P19)

I used to think 'I wished the thing had never fallen on me!' but now I think I've got to that stage where I've passed that and I think I have just slipped, probably slipped back into normality if, um, I, I sort of lived with this problem and that's then become me, so maybe you adapt. (P21)

Participants tried to return to usual activities but recovery was characterised by ups and downs. Some felt their recovery was delayed by further injuries described by participants as 'stress fractures', 'snapped (torn)

hamstrings' (muscles at the back of the thigh), or injury to their 'iliotibial bands' (connective tissue running from the pelvis to the knee). Only a small number of participants reported receiving enough physiotherapy after their injury. Those who received physiotherapy (usually privately) described a more purposeful recovery with fewer examples of setbacks and less uncertainty.

> I think she took it upon herself to get me back to where I wanted to be and I think she listened to what I wanted out of life. (P18)

Some participants expressed gratitude to be alive or to have avoided amputation. They tried to locate meaning in what happened to them.

> I went back and met another patient and his family and I hope that I gave them a little bit more hope. … It was good for me as well for my rehabilitation to feel that it wasn't all for nothing. (P18)

For these participants, they were able to find meaning in their injury, and so to incorporate it into their biography, or their story of themselves. Participants engaged in 'body work' in order to convey a sense of normality and provide the appearance of a non-injured body. However, often participants described struggling with the difference between others' expectations and the reality of their everyday experience.

> I think when people think you're doing so well physically they just think 'she's doing really well' and you almost don't want to turn around and say 'well actually I'm really struggling with this, that and the other' because I just don't want to disappoint people or turn it into a negative thing. (P14)

In recreating me, the participants were hindered by their injured body and a lack of supportive therapies but tried to integrate their past and present self-identity. New opportunities and meaningful ways of living that were beneficial could be found. There was a strong desire to appear normal and participants struggled to be themselves.

## DISCUSSION

This qualitative study adds to recent research on the patient experience of open fractures of the lower limb in acute care[3 9] and postinjury[4] by identifying the ongoing recovery undertaken by patients to process the impact of injury on their sense of being disempowered, being changed and being myself. We have specifically focussed on the longer term (2–4-years postinjury) impact of these injuries and especially in those patients who were identified as having injuries on the 'severe' end of the spectrum. We found that the concept of embodied vulnerability,[3 9] and endurance in early recovery,[8] clearly identified in acute care, also extends into the longer term as participants struggle to recover, processing their loss, working to negotiate how they live and integrate changes within their self-identity. Our findings indicate that longer term clinical support is required to improve outcomes for mental health, function, management of pain and living with disability in patients with major trauma to the lower limbs.

There were some limitations to our study. The sample was limited to those taking part in the UK WOLLF trial and participants may have different attributes to those who decline to take part in a clinical trial and be interviewed. Those agreeing to take part in an interview up to 4 years after injury may be less likely to consider themselves as recovered, thus be more representative of those with chronic ongoing problems. The sample was also not ethnically diverse and sampling at specific time points during recovery may have increased the transferability of the study findings to other populations. However, the UK WOLLF sample were considered to be representative of the general population with severe open fractures.[2] Our sample was purposive within the UK WOLLF population and saturation of data was achieved. Four patients with similar injuries (UK WOLLF PPI group) felt their personal experience resonated with the findings of this study; however, they also highlighted the joy of recovery, gratefulness to staff for their care, the similarity of the findings to, but also the difference from, their individual experience. Despite these limitations, this study provides evidence that patients who suffered the more severe injuries may benefit from enhanced clinical attention that focusses on distress, uncertainty, fragility and body image over a prolonged period of time. Psychological distress[6 13] is present in this group and may be linked to ongoing disability.[14] The continued uncertainty that patients feel reflects elements of a chaos narrative,[15] where loss of control means it is hard to find meaning in daily life. In our study despair, noted in recovery from trauma,[16] was expressed as suicidal thoughts. Their perceptions of the body as fragile and weak suggest a state of disembodiment[17 18] where dysfunction highlights the loss of normal taken for granted ways of being. Ongoing disruption to body image is noted in studies of stigma and disfigurement[19] and in societal pressure to appear recovered[20] and attractive.[21] In this study, amputation was considered as an alternative solution to continued challenges and dissociation from the body occurred, as in other specialities.[22] Patients' progression towards integration of physical change and self-identity was hindered by reminders of their injury, such as persistent pain, as their unreliable body[23] was unable to achieve the intended activity. This process is noted in chronic illness[24] and major trauma patients 3–6 months postinjury.[25] Support that enables patients to feel a greater sense of empowerment and integrate bodily changes within their self-identity may be helpful. Research should therefore focus on developing and testing effective interventions that provide longer term support and self-management within a holistic rehabilitation plan.

**Acknowledgements** The authors would like to thank all the patients who generously gave their time and energy to support this study. In addition, we would like to thank the PPI participants and staff who facilitated aspects of this study particularly Chris Bouse.

**Contributors** SR and ET contributed equally to this paper. The design, data collection, analysis and drafting of the paper were jointly undertaken by SR and ET. As a team, JA, JB and MLC supported the project and were involved in discussion of the findings and the development of this paper.

**Funding** This work was supported by the UK National Institute for Health Research, Health Technology Assessment (HTA) Programme (project number 10/57/20) and was supported by the National Institute for Health Research (NIHR) Oxford Biomedical Centre.

**Competing interests** None declared.

**Patient consent for publication** Not required.

**Ethics approval** The project was given ethical approval by West Midlands Coventry and Warwickshire Research Ethics Committee (REC Reference: 12/WM/0001) in June 2016.

**Provenance and peer review** Not commissioned; externally peer reviewed.

**Data availability statement** No data are available.

**ORCID iDs**
Sophie Rees http://orcid.org/0000-0003-4399-2049
Elizabeth Tutton http://orcid.org/0000-0003-3973-360X

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
