## [Reviewer comments · BMJ Open]

ARTICLE DETAILS

TITLE (PROVISIONAL)	Patient experience of long term recovery after open fracture of the lower limb: A qualitative study using interviews in a community setting
AUTHORS	Rees, Sophie; Tutton, Elizabeth; Achten, Juul; Bruce, Julie; Costa, Matthew L

VERSION 1 – REVIEW

REVIEWER	Ryan Trickett Cardiff and Vale University Health Board, UK
REVIEW RETURNED	02-Jul-2019

GENERAL COMMENTS	This study adds to the growing qualitative literature following lower limb injury. These studies are important as they demonstrate the importance of issues that may seem irrelevant to clinicians. Perhaps more importantly, along with the quantitative research in this field, they demonstrate that recovery – in the truest sense - seems to evade most patients who suffer a severe open lower limb fracture. Furthermore, the need for additional care in terms of mental health, distress, fragility and body image is highlighted – areas which unfortunately rarely form part of the trauma team's armamentarium. I have a few minor points concerning the presentation of the manuscript and some of the methodology/results. Overall, I consider this an important contribution to the literature and should be suitable for publication considering the points below: Page 3 comments that most participants had a Gustilo grade of 3. Was this based on the original Gustilo and Anderson description (as referenced) or that described by Gustilo, Williams and Mendoza? Can the authors explain Table 1 which has a similar problem? The authors have listed Gustilo grading "3+". Does this reflect Gustilo grade 3A, 3B or 3C or some other iteration of the grading classification. The explanation of the UK WOLLF PPI group does not read well. Consider "Four patients with similar injuries who were members of the UK WOLLF PPI group, helped the researchers reflect on their interpretations during analysis." In the methods the authors describe the discussion of the findings with the broader research team. It would be useful to understand who was responsible for the coding and categorisation stages as this may reflect on the conclusions drawn. Was there agreement with the results from the broader research team? In the methodology the authors mention the additional insights being challenges with intimacy and gratefulness for care provided. It is not clear whether this reflects the thoughts of the study group or the four "PPI" participants who reviewed the researcher's interpretations. Mention is also made of "multidisciplinary staff"
--

	who found resonance with the study findings. Who were these staff and in what capacity have the findings been reviewed? When was saturation achieved? Did the researchers continue recruiting to confirm saturation? If the four PPI patients introduced 2 new insights of intimacy challenges and gratefulness, was saturation truly achieved? For the reader unfamiliar with qualitative research I think the hierarchy of the overarching theme... “key themes” ... and categories could be better demonstrated in the manuscript. I find the emboldened title of the overarching theme unnecessary. The “Being unable to move freely” category in table 2 is also described as “being able to move freely” in the prose. Page 11 – by “muscle graft” are the authors quoting a participant or describing soft tissue/muscle flaps used in the reconstructive surgery? Page 12 – the “injuries occurred from stress fractures to snapped hamstrings...” sentence seems out of context and not necessarily phrased appropriately for a medical text. Are these subsequent injuries that the participants experienced? Page 15 – That the recruited participants were also participants in a surgically based randomised control trial is an important limitation of this study. Arguably, whilst those patients participating in research (be that qualitative or quantitative) may be different to those who decline, patients willing to participate in a surgical RCT may be different again. Do the authors think that these patients reflect not only the demographics of their clinical workload (as discussed briefly on page 15 regarding ethnicity) but also the character attributes?
--	--

VERSION 1 – AUTHOR RESPONSE

Reviewer: 1

1) This study adds to the growing qualitative literature following lower limb injury. These studies are important as they demonstrate the importance of issues that may seem irrelevant to clinicians. Perhaps more importantly, along with the quantitative research in this field, they demonstrate that recovery ? in the truest sense - seems to evade most patients who suffer a severe open lower limb fracture. Furthermore, the need for additional care in terms of mental health, distress, fragility and body image is highlighted ? areas which unfortunately rarely form part of the trauma team?s armamentarium.

Thank you we appreciate your support for this research which builds on earlier work in this area.

2) I have a few minor points concerning the presentation of the manuscript and some of the methodology/results. Overall, I consider this an important contribution to the literature and should be suitable for publication considering the points below:

Page 3 comments that most participants had a Gustilo grade of 3. Was this based on the original Gustilo and Anderson description (as referenced) or that described by Gustilo, Williams and Mendoza? Can the authors explain Table 1 which has a similar problem? The authors have listed Gustilo grading ?3+?. Does this reflect Gustilo grade 3A, 3B or 3C or some other iteration of the grading classification.

Thank you, apologies for the confusion, the following has been corrected and several sentences added to provide clarity. P3 and 4.

A key eligibility criteria for the UK WOLLF trial was that at the end of the first surgical wound debridement the wound could not be closed. This was required for the trial, as the NPWT dressings under investigation cannot be applied to closed wounds, but does not fit clearly with the existing

classification systems. Only patients with a higher grade of open fracture were included where skin or muscle grafts are required. Taking this into account, we used three sub-divisions of the Gustilo and Anderson[10] classification, grade II, grade III or grade III with a vascular injury. The classification in table 1 has been clarified. P5.

2) The explanation of the UK WOLFF PPI group does not read well. Consider 'Four patients with similar injuries who were members of the UK WOLFF PPI group, helped the researchers reflect on their interpretations during analysis.'

Thank you, this phrase has been added to improve clarity. P5.

3) In the methods the authors describe the discussion of the findings with the broader research team. It would be useful to understand who was responsible for the coding and categorisation stages as this may reflect on the conclusions drawn. Was there agreement with the results from the broader research team?

Thank you, the following has been added. P5 and 6.

Analysis was undertaken by two researchers (SR and ET). P5.

The findings were shared with the broader research team. Differences in interpretation were discussed as part of the process of analysis but in general there was agreement about the nature and content of the three themes. P6.

4) In the methodology the authors mention the additional insights being challenges with intimacy and gratefulness for care provided. It is not clear whether this reflects the thoughts of the study group or the four 'PPI' participants who reviewed the researcher's interpretations. Mention is also made of 'multidisciplinary staff' who found resonance with the study findings. Who were these staff and in what capacity have the findings been reviewed?

Thank you, the four PPI participants were part of the UK WOLFF PPI group and placed emphasis on existing codes in relation to intimacy and gratefulness, items that were important to them. This has now been clarified in the text. Additional information about the staff group has been added to improve clarity. P6.

Saturation of themes and categories was achieved. This was indicated when no new themes or categories developed after 18 patients were interviewed. Interviewing continued to ensure the sample was purposeful and there was a range of codes in each category. Resonance with the findings was identified by four patients from the UK WOLFF PPI group. It was noted that they placed emphasis on existing codes that identified challenges with intimacy and gratefulness for care provided. A workshop including a range of multidisciplinary staff representing nursing, physiotherapy, trauma surgeons, plastic surgeons and psychology suggested the findings reflected their experience of listening to patients in clinical practice, although they held differing perspectives on the degree of importance of items and labelling of codes. For example they felt some codes could be drawn together under the broader labels of anxiety or depression.

5) When was saturation achieved? Did the researchers continue recruiting to confirm saturation? If the four PPI patients introduced 2 new insights of intimacy challenges and gratefulness, was saturation truly achieved?

Thank you, saturation is, we acknowledge, a contested term for which there are differing perspectives. We have clarified our approach in point 4 and hope this is now clear. Intimacy and gratefulness were present in the data and the group expanded on their experience in relation to these areas. This has been clarified in the text in point 4. P6.

6) For the reader unfamiliar with qualitative research I think the hierarchy of the overarching theme? 'key themes' ? and categories could be better demonstrated in the manuscript. I find the emboldened title of the overarching theme unnecessary.

Thank you, the table has been removed and definitions of themes and categories have been placed within the text throughout the findings.

The emboldened title has been removed. P6.

7) The 'Being unable to move freely?' category in table 2 is also described as 'being able to move freely?' in the prose.

Apologies, this has been corrected. P12.

8) Page 11 'by 'muscle graft' are the authors quoting a participant or describing soft tissue/muscle flaps used in the reconstructive surgery?

Thank you, this has been rephrased to clarify the participants' perspective. P12

Specific activities such as being able to kneel prevented them from moving as freely. Some participants felt the area where they had a soft tissue/muscle flap, used in reconstructive surgery, was heavy (notable when sleeping), or they experienced swelling and pain.

9) Page 12 'the 'injuries occurred from stress fractures to snapped hamstrings??' sentence seems out of context and not necessarily phrased appropriately for a medical text. Are these subsequent injuries that the participants experienced?

Thank you, this has been rephrased to reflect the participants' description of their subsequent injuries. It is their description and interpretation of these injuries rather than clinically confirmed diagnoses.

Participants tried to return to usual activities but recovery was characterised by ups and downs. Some felt their recovery was delayed by further injuries that participants described as stress fractures, snapped (torn) hamstrings (muscles at the back of the thigh), or injury to their iliotibial bands (connective tissue running from the pelvis to the knee). P15.

10) Page 15 'That the recruited participants were also participants in a surgically based randomised control trial is an important limitation of this study. Arguably, whilst those patients participating in research (be that qualitative or quantitative) may be different to those who decline, patients willing to participate in a surgical RCT may be different again.

Do the authors think that these patients reflect not only the demographics of their clinical workload (as discussed briefly on page 15 regarding ethnicity) but also the character attributes?

We agree that those taking part in research in general or a surgically based randomised controlled trial, and those declining to take part in a trial may have different attributes. The professional group found the patient experience familiar to that observed in clinical practice and this has been clarified in point 4, P6. There is more information about the comparison between those who were consented and non-consented in the UK WOLLF monograph publication which is referenced.

The following has been added to the text. P16.

The sample was limited to those taking part in the UK WOLLF trial and participants may have different attributes to those who decline to take part in a clinical trial and be interviewed. Those agreeing to take part in an interview up to four years after injury may be less likely to consider themselves as recovered, thus be more representative of those with chronic ongoing problems.

And later,

However the UK WOLLF sample were considered to be representative of the general population with severe open fractures.[2]

Thank you for your feedback which has helped to improve this manuscript.

VERSION 2 – REVIEW

REVIEWER	Ryan Trickett Cardiff and Vale University Health Board Previous publication of qualitative findings in the first 2 years following open tibial fracture.
REVIEW RETURNED	28-Aug-2019

GENERAL COMMENTS	I thank the authors for their responses to my previous points. All have been addressed. Regarding the fracture classification - I understand the rationale and requirement for including only the more severe injuries, but I think it would be clearer for readers if the demographics stated the numbers as either grade II and III(inclusive of a/b/c; or II, IIIa, IIIb and IIIc. Personally, I think that the brief descriptor of participants after the quotes in the results adds context and could be retained. If not retaining them then “P02, 69 years, female...” should be deleted from page 11. I think the additional injuries described (in patient terms) on page 15 could be placed into quotation marks, or as a full indented quotation. If these changes are considered I feel this paper is suitable for publication.
--

VERSION 2 – AUTHOR RESPONSE

1) Regarding the fracture classification - I understand the rationale and requirement for including only the more severe injuries, but I think it would be clearer for readers if the demographics stated the numbers as either grade II and III(inclusive of a/b/c; or II, IIIa, IIIb and IIIc.

Thank you, this has been added to the text and table. P4/5

2) Personally, I think that the brief descriptor of participants after the quotes in the results adds context and could be retained. If not retaining them then “P02, 69 years, female...” should be deleted from page 11.

The editorial process has required that we remove descriptors as they contain information such as age and gender. Thank you, I have removed the descriptor for P02.

3) I think the additional injuries described (in patient terms) on page 15 could be placed into quotation marks, or as a full indented quotation.

Thank you, quotation marks have been used. P13